# VISUAL OBJECT-CENTRIC COUNTERFACTUAL EXPLANATIONS

## ABSTRACT

Generating visually coherent and realistic counterfactual explanations is essential for understanding discriminative visual models. Existing methods often modify images at the pixel-level or within holistic latent spaces, leading to entangled changes that obscure the precise factors influencing model decisions. To address this, we introduce a novel object-centric method for visual counterfactual explanations. Our approach decomposes input images into distinct object-centric latent slots and leverages model's gradients to guide a reverse diffusion process conditioned on these slots. To maintain realism, we propose a Gaussian Mixture Model (GMM)-based regularizer that constrains counterfactuals to remain within the distribution of plausible object states, preventing unrealistic generations. Experiments on three datasets and a user study demonstrate that our object-centric approach yields significantly more interpretable and realistic counterfactuals compared to state-of-the-art baselines. Moreover, our approach shows strong generalization: when trained solely on FFHQ dataset, it successfully generates coherent counterfactual explanations on unseen CelebA-HQ data. Overall, our approach substantially advances visual counterfactual explanations by offering explicit object-level interpretability and improved quality of generation.

## 1 INTRODUCTION

The pursuit of explainability and interpretability in machine learning has become essential as complex models increasingly permeate critical decision-making processes (Goodman & Flaxman, 2017; Doshi-Velez & Kim, 2017; Rudin, 2019; Alufaisan et al., 2021; Pedreschi et al., 2019). Among various methods developed to enhance model transparency, counterfactual explanations have gained particular prominence due to their intuitive appeal and actionable insights (Wachter et al., 2017; Goyal et al., 2019). A counterfactual explanation provides a minimally altered version of an input instance that shifts the model's prediction toward a specified outcome, thereby clarifying the causal relationship between input attributes and model decisions. For example, subtly modifying visual attributes of an object to alter a classification outcome from "cat" to "dog" offers human-interpretable evidence of features driving the model's behavior (Goyal et al., 2019; Chang et al., 2018; Kim et al., 2021). Crucially, effective visual counterfactuals require targeted and semantically meaningful modifications—such as changing object-specific characteristics rather than background details—thus enabling users to verify whether model decisions align with human intuition (Hendricks et al., 2018). By emphasizing localized and interpretable alterations, counterfactual explanations not only enhance model transparency but also facilitate rigorous evaluation of the semantic grounding and generalizability of visual classifiers (Smyth & Keane, 2022).

Despite the notable progress, generating high-quality visual counterfactual explanations remains challenging. Classical counterfactual techniques predominantly focus on tabular data, where individual features (e.g. age, income) are inherently interpretable and straightforward to manipulate (Wachter et al., 2017; Molnar, 2020; Romashov et al., 2022). In contrast, visual data, represented at the pixel level, lacks direct interpretability due to the high dimensionality and entangled semantics inherent in raw image space (Goyal et al., 2019; Hendricks et al., 2018). Recent work addresses this limitation by generating counterfactuals in learned latent feature spaces, where abstract representations better isolate semantically meaningful changes (Rodriguez et al., 2021; Lang et al., 2021; Kirilenko et al., 2024). Building upon this insight, we propose to further enhance interpretability by generating visual counterfactuals within an explicitly object-centric latent space, represented by

slot-based decompositions (Locatello et al., 2020). Slot-based representations naturally disentangle object-level semantics, enabling highly targeted and intuitive modifications for counterfactual generation (Greff et al., 2020).

Building upon a legacy of object-centric visual cognition (Fukushima, 1980; Greff et al., 2020), we address a question left open by prior counterfactual work: *Can a semantically grounded, unsupervised object-centric generative model produce better counterfactual explanations than pixel-level or holistic latent approaches?*

Existing methods typically entangle foreground and background features, leading to counterfactuals that flip the prediction but also alter irrelevant parts of the image, making it unclear which change was causally responsible. We close this gap by introducing a novel object-centric framework for visual counterfactual explanations.

Our contributions are:

- **Object-centric counterfactual framework:** Our approach operates on *slot-based representations*, where each slot encodes a distinct object. These representations are learned in an unsupervised manner and provide a structured, interpretable foundation for semantic counterfactual intervention.

- **Classifier-guided slot diffusion with GMM realism prior:** We modify standard classifier-guided diffusion, introducing a Gaussian Mixture Model (GMM) prior that constrains edits to remain within the distribution of plausible object states. This yields a *targeted and semantically grounded generator* of counterfactual explanations.

- **Comprehensive evaluation:** On ClevrTex, FFHQ, and CelebA-HQ datasets, our method outperforms state-of-the-art approaches for visual counterfactual explanations.

- **User study and ablations:** Human evaluations show our explanations are rated as more *meaningful, subtle, and realistic*. Ablation studies confirm the contribution of key architectural components to overall performance.

- **Cross-dataset generalization:** Despite training only on FFHQ, our model generates high-quality counterfactuals on unseen CelebA-HQ images substantially outperforming non-slot-based baselines.

## 2 BACKGROUND AND RELATED WORKS

### 2.1 COUNTERFACTUAL EXPLANATIONS

Counterfactual explanations aim to answer: *"What is the smallest meaningful change to a given input that would cause a **specified model** to predict a desired outcome?"* (Wachter et al., 2017; Goyal et al., 2019; Poyiadzi et al., 2020; Kenny & Keane, 2021). Formally, for a predictor $f : \mathcal{X} \to \mathcal{Y}$, a query $\mathbf{x}$ with prediction $y = f(\mathbf{x})$, and a target label $y^*$, a counterfactual $\mathbf{x}_{cf}$ is often defined as

$$\mathbf{x}_{cf} = \arg\min_{\mathbf{x}'} d(\mathbf{x}, \mathbf{x}') \quad \text{s.t.} \quad f(\mathbf{x}') = y^*, \tag{1}$$

where $d(\cdot, \cdot)$ is a distance metric. Three desiderata are typically enforced: (i) **validity** ($f(\mathbf{x}_{cf}) = y^*$), (ii) **minimality/closeness** (minimal distance $d(\mathbf{x}, \mathbf{x}_{cf})$), and (iii) **plausibility/realism** (the change should live on or near the data manifold and be semantically meaningful) (Rudin, 2019; Molnar, 2020).

**Not just conditional generation.** Sampling from a conditional generator $p(\mathbf{x} \mid y^*)$ solves a different problem: it produces *any* plausible sample with attribute $y^*$, irrespective of the specific $\mathbf{x}$ or the behavior of $f$. Counterfactuals are *instance-specific* (they stay close to $\mathbf{x}$) and *model-specific* (success is defined by $f$, not by the generator). Merely drawing a realistic face with glasses does not explain why $f$ misclassified *this* face without glasses, nor what minimal edit would flip $f$'s decision. Equation 1 captures this distinction, i.e., it is an optimization process anchored at $\mathbf{x}$ and constrained by $f$, not a draw from $p(\mathbf{x} \mid y^*)$.

**Visual counterfactuals.** While Eq.1 mirrors the optimization used in adversarial attacks, adversarial examples are agnostic to human meaning and realism (Freiesleben, 2022; Pawelczyk et al., 2022); counterfactuals, in contrast, must produce interpretable, plausible edits, this extra burden becomes

acute for images. Pixel space is high-dimensional and semantically entangled, so naive perturbations either create implausible edits or spill them across foreground and background, obscuring causal factors and often drifting off-manifold. This makes realism and locality constraints substantially harder than in tabular settings (Goyal et al., 2019; Hendricks et al., 2018).

Prior works address this by performing constrained manipulations in the learned latent space to carve out semantically meaningful directions (Rodriguez et al., 2021; Lang et al., 2021; Kirilenko et al., 2024) or they limit targeted regions using saliency maps (Samadi et al., 2023; 2024) or use information about scene composition either through semantic segmentation (Jacob et al., 2022) or object parameters (Zemni et al., 2023). We follow the same direction: instead of holistic latents, we operate on an explicitly object-centric compositional representations, improving locality and overall quality of counterfactual edits.

## 2.2 DIFFUSION MODELS

Denoising Diffusion Probabilistic Models (DDPMs) (Ho et al., 2020; Nichol & Dhariwal, 2021) learn to generate data by reversing a noising process. The forward diffusion process adds Gaussian noise

$$q(\mathbf{x}_t \mid \mathbf{x}_{t-1}) = \mathcal{N}\big(\sqrt{1 - \beta_t}\, \mathbf{x}_{t-1},\, \beta_t \mathbf{I}\big), \tag{2}$$

which implies the closed form

$$q(\mathbf{x}_t \mid \mathbf{x}_0) = \mathcal{N}\big(\sqrt{\bar{\alpha}_t}\, \mathbf{x}_0,\, (1 - \bar{\alpha}_t)\mathbf{I}\big), \tag{3}$$

with $\bar{\alpha}_t = \prod_{j=1}^{t}(1 - \beta_j)$. A U-Net $\epsilon_\theta$ is trained to predict the injected noise via the usual objective

$$\mathcal{L} = \mathbb{E}_{\mathbf{x}_0, t, \epsilon}\left[\left\|\epsilon - \epsilon_\theta(\mathbf{x}_t, t)\right\|^2\right], \tag{4}$$

which enables the reverse diffusion process $p_\theta(\mathbf{x}_{t-1}, \mathbf{x}_t)$.

In this framework it is possible to incorporate external classifier signal $p_\phi(y|\mathbf{x})$ into $p_{\theta,\phi}(\mathbf{x}_{t-1}|\mathbf{x}_t, y)$ (Dhariwal & Nichol, 2021). Classifier-guided diffusion modifies the reverse transition probabilities as follows:

$$p_{\theta,\phi}(\mathbf{x}_{t-1}|\mathbf{x}_t, y) \propto p_\theta(\mathbf{x}_{t-1}|\mathbf{x}_t)\, p_\phi(y|\mathbf{x}_{t-1}). \tag{5}$$

This feature made diffusion models popular for producing visual counterfactuals. DVCE (Augustin et al., 2022) and DiME (Jeanneret et al., 2022) apply classifier guidance with additional regularizers to steer edits toward counterfactuals, while ACE (Jeanneret et al., 2023) integrates an inpainting stage to refine adversarial perturbations into more realistic image changes. Beyond these, DiG-IN (Augustin et al., 2024) leverages diffusion counterfactuals to widely explore classifier failure modes, and ECED (Luu et al., 2025) incorporates latent diffusion with saliency maps to localize edits to semantically relevant regions.

## 2.3 OBJECT-CENTRIC LEARNING

The field of object-centric learning draws inspiration from the object-oriented nature of human visual perception, where our visual system instinctively decomposes scenes into discrete entities and familiar structures Spelke (1990); Grill-Spector (2003). Motivated by this cognitive ability, object-centric models typically learn to map an input image $\mathbf{x} \in \mathbb{R}^{H \times W \times C}$ into an unordered set of latent vectors, commonly referred to as slots $\mathbf{s} = \{\mathbf{s}_1, \mathbf{s}_2, \dots, \mathbf{s}_K\}$, in an unsupervised manner, where each slot $\mathbf{s}_k \in \mathbb{R}^d$ encodes the properties of a distinct object or entity observed in the original input $\mathbf{x}$.

Recent advancements in object-centric learning have led to numerous influential approaches Eslami et al. (2016); Greff et al. (2017); Burgess et al. (2019); Engelcke et al. (2019) demonstrating significant potential in decomposing complex visual scenes into interpretable components. Slot Attention (SA) (Locatello et al., 2020) has emerged as the most prominent method in object-centric learning, widely adopted for its simplicity and effectiveness (Lee et al., 2024; Li et al., 2021; Kipf et al., 2021; Elsayed et al., 2022; Singh et al., 2021; Yoon et al., 2023). SA iteratively maps a flattened distributed feature representation $\mathbf{z} \in \mathbb{R}^{N \times D}$ from input images $\mathbf{x}$ into a set of randomly initialized slots $\mathbf{s}$. These slots compete through dot-product attention, assigning portions of the input to specific slots in a manner reminiscent of soft K-means clustering combined with Gated Recurrent Unit (GRU) (Cho et al., 2014) and Multi-Layer Perceptron (MLP) (Rumelhart et al., 1986) updates.

# 3 METHOD

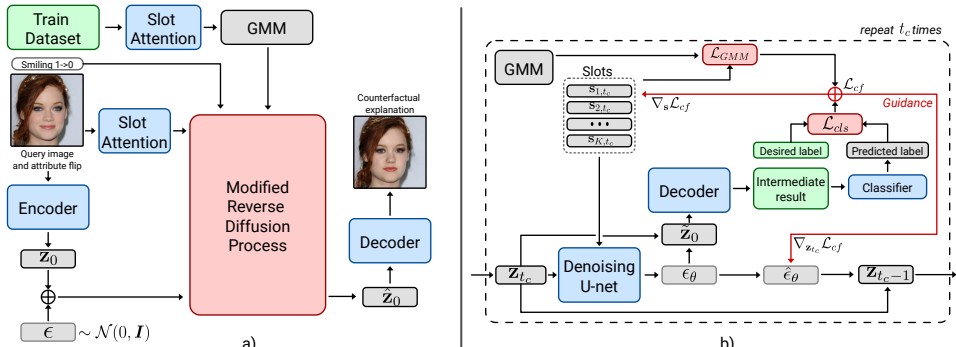

Figure 1: Illustration of the proposed approach. *(a) High-level pipeline.* A query image $\mathbf{x}$ is encoded into a latent vector $\mathbf{z}_0$, while Slot-Attention extracts slots $\mathbf{s}$. We perform a $t_c$ forward diffusion step using $q(\mathbf{z}_{t_c} \mid \mathbf{z}_0)$, and then invoke a slot-conditioned reverse-diffusion process that is further guided by a GMM fitted to training-set slots and a desired attribute flip. The refined latent $\tilde{\mathbf{z}}_0$ is decoded, yielding the counterfactual. *(b) Single reverse-diffusion step.* The noisy latent $\mathbf{z}_{t_c}$ is passed through a denoising U-Net to predict $\epsilon_0$ and an intermediate latent $\hat{\mathbf{z}}_0$. A classifier provides $\mathcal{L}_{\text{cls}}$, and the GMM supplies a plausibility term $\mathcal{L}_{\text{GMM}}$. The combined guidance adjusts the slots $\mathbf{s}_{t_c}$ and the noise estimate from $\epsilon_\theta$ to $\hat{\epsilon}_\theta$, which is then used to compute the next $\mathbf{z}_{t_c-1}$.

In this section, we introduce *Visual Object-Centric Counterfactual Explanations (VOCCE)*, our proposed approach for generating object-centric visual counterfactuals. VOCCE operates by modifying latent slot representations within a diffusion-based generative framework, enabling the creation of targeted, semantically rich counterfactual examples. Our method comprises four key components:

1. **Latent Slot Diffusion**, a state-of-the-art object-centric generative model.
2. **Classifier Guidance**, which steers the diffusion process toward desired label by adjusting the noise prediction.
3. **Gradient-Based Slot Updates**, where classifier gradients modify slot representations.
4. **GMM-Based Regularization**, which ensures updated slots remain within the distribution of plausible states.

## 3.1 LATENT SLOT DIFFUSION

We adopt LSD as the core backbone of our counterfactual generation pipeline. Given a query sample $\mathbf{x}_q \in \mathbb{R}^{H \times W \times C}$ and a target discriminative model $f : \mathbb{R}^{H \times W \times C} \to \mathbb{R}^L$, our goal is to produce a counterfactual for a desired label $\mathbf{y}^* \in \mathbb{R}^L$, $\mathbf{y}^* \neq f(\mathbf{x}_q)$. We begin by encoding $\mathbf{x}_q$ into the latent space, $\mathbf{z}_0 = \text{Enc}(\mathbf{x}_q)$, and extracting slot representations, $\mathbf{s} = \text{SA}(\mathbf{x}_q)$. Next, we partially noise $\mathbf{z}_0$ to obtain $\mathbf{z}_{t_c} \sim q(\mathbf{z}_{t_c} \mid \mathbf{z}_0)$ at a chosen timestep $t_c < T$.

We then perform reverse diffusion using the trained denoising model $\epsilon_\theta(\mathbf{z}_t, t, \mathbf{s})$. During this reverse process, we leverage gradients from the target model $f$ to update both the predicted noise $\hat{\epsilon}_\theta$ and the slot representations $\mathbf{s}$, thus steering the diffusion towards the desired label $\mathbf{y}^*$. Once the process reaches $\hat{\mathbf{z}}_0$, we decode it to the image space, yielding the final counterfactual $\mathbf{x}_{cf} = \text{Dec}(\hat{\mathbf{z}}_0)$.

## 3.2 CLASSIFIER GUIDANCE

Following state-of-the-art diffusion-based methods for visual counterfactual explanations (Augustin et al., 2022; Jeanneret et al., 2022; 2023), we incorporate classifier guidance by leveraging the gradient of a classifier loss $\nabla_{\mathbf{z}_t} \mathcal{L}_{\text{cls}}(\mathbf{y}^*, f(\tilde{\mathbf{x}}))$, where $\tilde{\mathbf{x}} = \text{Dec}(\tilde{\mathbf{z}}_0)$. The term $\tilde{\mathbf{z}}_0$ is an approximate reconstruction of $\mathbf{z}_0$ from the noisy sample $\mathbf{z}_t$, derived from Eq.3 via

$$\tilde{\mathbf{z}}_0 = \frac{1}{\sqrt{\bar{\alpha}_t}}\Big(\mathbf{z}_t - \epsilon_\theta(\mathbf{z}_t, t, \mathbf{s})\sqrt{1 - \bar{\alpha}_t}\Big). \tag{6}$$

To guide the diffusion process towards $\mathbf{y}^*$, we modify the predicted noise $\epsilon_\theta$ using the classifier gradient. Specifically, the guided noise approximation $\hat{\epsilon}_\theta$ becomes

$$\hat{\epsilon}_\theta \;=\; \epsilon_\theta \;-\; \sqrt{1 - \bar{\alpha}_t}\,\nabla_{\mathbf{z}_t}\,\mathcal{L}_{\text{cls}}\Big(\mathbf{y}^*,\, f(\tilde{\mathbf{x}})\Big). \tag{7}$$

By backpropagating through the approximate reconstruction $\tilde{\mathbf{z}}_0$ and the decoder $\text{Dec}(\cdot)$, the classifier's gradient encourages the generated sample to align with the target label $\mathbf{y}^*$.

### 3.3 GMM-BASED REGULARIZATION

To prevent slot modifications from drifting into out-of-distribution regions and to ensure that the resulting counterfactuals remain plausible, we introduce a **GMM-based regularization**. Prior to counterfactual generation, we collect slot representations from the training data samples and fit a Gaussian Mixture Model (GMM) with $M$ components:

$$p(\mathbf{s}) \;=\; \sum_{m=1}^{M} \pi_m\,\mathcal{N}\big(\mathbf{s};\, \boldsymbol{\mu}_m,\, \boldsymbol{\Sigma}_m\big). \tag{8}$$

Previous works have shown that clustering in the slot space yields a semantically meaningful grouping (Singh et al., 2021; Jiang et al., 2023; Kirilenko et al., 2023).

By modeling the distribution of slots, we can better constrain the counterfactual generation process. During inference, we use the learned GMM parameters $\{\boldsymbol{\mu}_m, \boldsymbol{\Sigma}_m\}$ to enforce a high likelihood under this prior. Specifically, we introduce a penalty term:

$$\mathcal{L}_{\text{GMM}} \;=\; -\sum_{i=1}^{K} \log p(\mathbf{s}_i), \quad \mathcal{L}_{\text{cf}} \;=\; \mathcal{L}_{\text{cls}} \;+\; \gamma\,\mathcal{L}_{\text{GMM}}, \tag{9}$$

where $\gamma$ balances classifier-driven modifications and realism constraints. $\mathcal{L}_{\text{GMM}}$ penalizes slot configurations that deviate significantly from the training distribution. In our experiments, we set $\gamma = 0.1$, finding this value effective in preserving the plausibility of the generated counterfactuals while encouraging them to correspond to the desired label.

### 3.4 GRADIENT-BASED SLOT UPDATES

Beyond the typical classifier-guided noise adjustments in diffusion models, we note that the approximate reconstruction $\tilde{\mathbf{x}}$ also depends on the slot representations $\mathbf{s}$. Since

$$\tilde{\mathbf{x}} = \text{Dec}\big(\tilde{\mathbf{z}}_0(\mathbf{z}_t,\, \epsilon_\theta(\mathbf{z}_t,\, t,\, \mathbf{s}))\big), \tag{10}$$

we can backpropagate through both the decoder $\text{Dec}(\cdot)$ and the denoising network $\epsilon_\theta$ to update $\mathbf{s}$ itself. This additional gradient pathway enables more direct manipulation of the latent factors responsible for object- or region-level changes in the generated counterfactual.

In each reverse diffusion step, we update $\mathbf{s}$ by descending its gradient:

$$\mathbf{s}_{t-1} \;\leftarrow\; \mathbf{s}_t \;-\; \eta_s\,\nabla_{\mathbf{s}}\mathcal{L}_{\text{cf}}\Big(\mathbf{y}^*,\, f(\tilde{\mathbf{x}})\Big), \tag{11}$$

where $\eta_s$ is a step size for the slot update.

Empirically, we observe that slot-level guidance alone already outperforms classifier-guidance–based baselines, producing more localized better counterfactuals; when combined with classifier-based noise adjustments, it yields further improvements.

## 4 EXPERIMENTAL SETUP

### 4.1 DATA

In our experiments we use three datasets: **ClevrTex** (30k images of simple scenes with various objects), **FFHQ** (70k images of human faces), and **CelebA-HQ** (30k high-resolution face images). ClevrTex provides detailed object-level annotations (shape, size, texture) for each object in a scene;

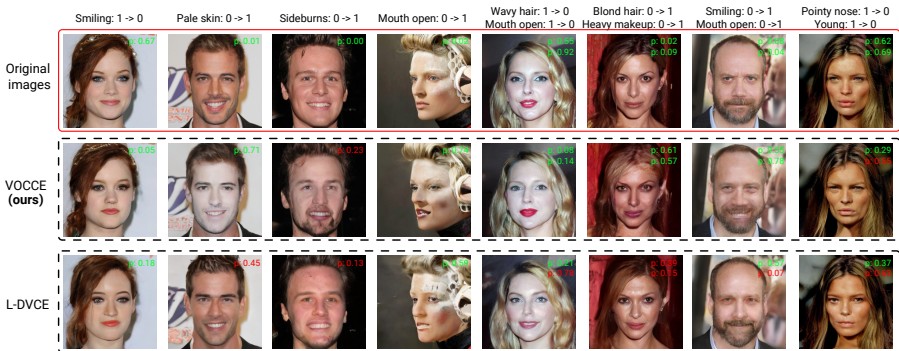

Figure 2: Random examples of attribute manipulations on CelebA-HQ using our proposed VOCCE approach and L-DVCE at $t_c = 250$. First four columns represents a distinct binary attribute shift (e.g., "Smiling: $1 \rightarrow 0$"), and the others represent two binary attributes shifts. The text "p: ..." corresponds to a classifier's predicted probability of the attribute; green indicates the prediction matches the target attribute, while red indicates a mismatch. VOCCE consistently demonstrates higher fidelity to the desired attribute modification and better preservation of other facial features compared to L-DVCE.

we create counterfactual targets by randomly selecting an object and flipping one of its attributes (e.g., shape, size, or texture). CelebA-HQ contains 40 binary facial attributes (*Smiling*, *Glasses*, *Young*, etc.), and we generate counterfactual labels by randomly choosing an attribute to invert. Since FFHQ does not include labels, we use it solely to augment training for models that are then tested on CelebA-HQ. All datasets are split into 80% training, 10% validation, and 10% test sets, we use 256×256 resolution images for both training and inference.

## 4.2 MODELS

**Backbone generative model.** We adopt LSD as our primary backbone and replicate the U-Net architecture along with all training hyperparameters from (Jiang et al., 2023). All generative models are trained for 300,000 iterations using the Adam optimizer with learning rate $3 \times 10^{-4}$ and a batch size of 64. Detailed model configs are included in the code appendix for reproducibility.

To regularize slot updates, we fit a GMM to the training-set slots. Following prior work (Jiang et al., 2023; Kirilenko et al., 2023), we use $M{=}18$ components for FFHQ and CelebA-HQ, and $M{=}5$ for ClevrTex, as these values have been shown to yield semantically coherent slot clusters.

For CelebA attribute classification, we fine-tune a ResNet-18 pretrained on ImageNet using binary cross-entropy loss across the 40 attributes. For ClevrTex, we train a ResNet-18 with three separate heads (for shape, size, and texture), applying the Hungarian algorithm for object matching following (Locatello et al., 2020). The resulting classifiers achieve mean average precision (mAP) scores of 0.87 on CelebA and 0.82 on ClevrTex.

**Baselines.** Two diffusion-based methods closely related to our approach are Diffusion Visual Counterfactual Explanations (DVCE) (Augustin et al., 2022) and Diffusion Models for Counterfactual Explanations (DiME) (Jeanneret et al., 2022). Both rely on reverse diffusion, but differ in how classifier evaluations are handled during denoising. DiME performs multiple unconditional denoising steps per conditional iteration, introducing significant computational overhead. In contrast, DVCE, like our method, uses a single-step approximation (see Eq. 6), making it more efficient and more comparable to our VOCCE method.

We adopt DVCE as a baseline and adapt its modification for a fair comparison. The original DVCE operates in pixel space, while we use a pretrained VAE (Esser et al., 2021; Rombach et al., 2022). We refer to this adapted version as *Latent-DVCE (L-DVCE)*. We also compare to FastDiME Weng et al. (2024), a modified version of DiME with single step denoising from Eq. 6 and self-optimized masking, in out experiments we find FastDiME to perform better than original DiME and use Fast-DiME as a baseline. All diffusion-base approaches share the same U-Net backbone, differing only in

Table 1: Performance of visual counterfactual-explanation methods on ClevrTex and CelebA-HQ (mean±std for 3 seeds). Best values highlighted with **bold**, second best are underscored. Proposed **VOCCE** achieves the best closeness, locality, and realism.

| DATASET | METHOD | CLOSENESS | | VALIDITY | | LOCALITY | REALISM |
|---|---|---|---|---|---|---|---|
| | | $l_2 \downarrow$ | LPIPS $\downarrow$ | Succ. (%) $\uparrow$ | Fail. (%) $\downarrow$ | ELS (%) $\uparrow$ | sFID $\downarrow$ |
| ClevrTex | **VOCCE**$_{t_c=250}$ | **19±1.3** | **0.15±0.01** | 85.3 ± 1.0 | **11.9±0.8** | **82.4±1.1** | **25.5±1.4** |
| | DVCE$_{t_c=250}$ | 24±1.8 | 0.20±0.01 | 82.0±1.3 | 14.2±0.9 | 63.0±1.4 | 31.0±1.7 |
| | L-DVCE$_{t_c=250}$ | 21±1.5 | 0.16±0.01 | 84.8±1.2 | 13.3±0.9 | 68.1±1.3 | 27.6±1.5 |
| | FastDiME$_{t_c=250}$ | 23±1.7 | 0.19±0.01 | 81.5±1.3 | 14.6±0.9 | 72.7±1.4 | 31.8±1.8 |
| | OCTET | 41±2.0 | 0.21±0.02 | 79.2±1.3 | 16.6±1.1 | 62.0±1.8 | 48.0±2.6 |
| | DiVE | 82±4.2 | 0.50±0.04 | **90.2±1.1** | 15.8±0.9 | 38.7±1.9 | 124.9±6.0 |
| CelebA-HQ | **VOCCE**$_{t_c=250}$ | **44±2.5** | **0.22±0.01** | 86.8±1.1 | 15.6±1.0 | **71.2±1.6** | **35.4±1.7** |
| | DVCE$_{t_c=250}$ | 52±3.0 | 0.30±0.02 | 67.1±1.6 | 16.0±1.1 | 52.4±1.7 | 40.3±2.1 |
| | L-DVCE$_{t_c=250}$ | 49±2.8 | 0.28±0.02 | 69.0±1.5 | **14.2±1.0** | 56.8±1.6 | 39.7±2.1 |
| | FastDiME$_{t_c=250}$ | 51±2.9 | 0.31±0.02 | 66.8±1.5 | 16.3±1.0 | 60.5±1.7 | 41.1±2.2 |
| | OCTET | 60±3.4 | 0.36±0.03 | 74.0±1.6 | 15.7±1.1 | 54.0±2.0 | 60.0±3.0 |
| | DiVE | 79±4.0 | 0.62±0.05 | **95.1±0.9** | 16.9±1.0 | 34.3±1.8 | 159.0±8.0 |

that VOCCE conditions on slot representations **s** via Cross-Attention, while L-DVCE and FastDiME replaces this with Self-Attention (Vaswani et al., 2017). We use DDIM sampling (Song et al., 2020) for counterfactual generation, using 200 total denoising steps. Given $t_c = 250$, this corresponds to 50 effective denoising steps.

To benchmark against non-diffusion approaches, we include DiVE (Rodriguez et al., 2021), a strong VAE-based counterfactual method, and OCTET (Zemni et al., 2023), an approach that leverages a GAN backbone developed for compositional scene generation. We scale all models hyperparameter to match the trainable parameter count of our diffusion-based models.

## 5 EXPERIMENTS

In this section, we evaluate our proposed VOCCE on both synthetic (ClevrTex) and real-world (CelebA-HQ) datasets, comparing against the baselines. We then examine generalization when training exclusively on FFHQ and testing on CelebA-HQ, and conduct ablations on the influence of the guidance signals and GMM regularization.

### 5.1 QUANTITATIVE METRICS

We measure the *closeness* of counterfactuals to corresponding queries using $l_2$ norm in the latent space produced by pretrained VAE from (Rombach et al., 2022), and the perceptual LPIPS metric (Zhang et al., 2018) computed from the actual images using the AlexNet model (Krizhevsky et al., 2012). We report *validity* with success and failure rates, where success rate is a fraction of successful target attribute flips, and failure rate is a fraction of unintentionally changed attributes (ignoring entangled CelebA-HQ attributes, e.g. *Black hair* and *Blond hair*). Failure rate can also be considered as another measure of *closeness*, as we want our methods to flip target attributes exclusively. *Realism* is measured with sFID (Jeanneret et al., 2023), computed as the Fréchet Inception Distance (?Seitzer, 2020) between counterfactuals of one image set and an independent set of real images from the same dataset.

We also quantify whether edits are localized correctly with an *Edit Localization Score* (ELS) that measures the fraction of change localized in the semantically relevant region and aligned with the classifier. Given the original image **x** and the counterfactual $\mathbf{x}_{cf}$, we compute a per-pixel change map $\Delta = \text{Gauss}(|\mathbf{x} - \mathbf{x}_{cf}|_1)$ using $\ell_1$ difference followed by a small Gaussian blur to suppress pixel noise. $\mathcal{R}$ denote the target semantic region (e.g., the ground-truth object mask on ClevrTex or a part mask such as *mouth* or *hair* for corresponding target attributes from CelebA-HQ). We also intersect $\mathcal{R}$ with the top-20% support of Grad-CAM (Selvaraju et al., 2017) saliency maps **S** for the

Table 2: Ablation of slot-level guidance on CelebA-HQ at $t_c = 250$. We vary the slot guidance step size $\eta_s$ and the optional GMM regularization. The first row is a *reference* (full VOCCE with slot+noise guidance); all others are use *slot-only* updates to produce counterfactuals.

| METHOD | CLOSENESS | | VALIDITY | | LOCALITY | REALISM |
|---|---|---|---|---|---|---|
| | $l_2 \downarrow$ | LPIPS $\downarrow$ | Succ. (%) $\uparrow$ | Fail. (%) $\downarrow$ | ELS (%) $\uparrow$ | sFID $\downarrow$ |
| **VOCCE (slot+noise)** | **44** | **0.22** | **86.8** | 15.6 | **71.2** | **35.4** |
| Slot-only + GMM$_{\eta_s=1.0}$ | 45 | 0.24 | 86.1 | 14.1 | 70.6 | 36.1 |
| Slot-only + GMM$_{\eta_s=0.3}$ | 46 | 0.23 | 86.0 | **14.0** | 70.9 | 36.0 |
| Slot-only $-$ GMM$_{\eta_s=1.0}$ | 56 | 0.33 | 75.4 | 17.0 | 54.2 | 46.0 |
| Slot-only $-$ GMM$_{\eta_s=0.3}$ | 49 | 0.28 | 78.5 | 16.5 | 60.1 | 44.7 |

target class. The score is computed as

$$\text{ELS} \;=\; 100 \times \frac{\sum_{i \in \mathcal{R} \cap \Omega_\tau(\mathbf{S})} \Delta_i}{\sum_i \Delta_i}, \tag{12}$$

so higher values indicate edits that are both spatially localized and classifier-relevant.

Table 1 summarizes the performance of VOCCE and baselines on ClevrTex and CelebA-HQ. For each row we take the same 1k test images and randomly create 5 shared target counterfactual labels, which results in 5k counterfactuals per row. VOCCE consistently outperforms all baselines in terms of *closeness* (measured by $l_2$ and LPIPS), *locality*, and *realism*, establishing itself as the strongest method overall. Although DiVE achieves higher *validity*, it does so at the cost of significantly worse closeness, introducing greater distortions to the images, as expected from a non-diffusion-based baseline, and substantially higher sFID scores, indicating lower visual fidelity.

## 5.2 QUALITATIVE ANALYSIS

Figure 2 illustrates examples of counterfactual explanations on CelebA-HQ for both VOCCE and L-DVCE. First four columns displays a single binary attribute shift (e.g., "Smiling: $1 \rightarrow 0$") while the others 4 depict double attribute shift. VOCCE not only meets the target attribute more consistently (green probabilities) but also better preserves other facial details. In contrast, L-DVCE occasionally struggles with maintaining non-targeted features, which results in the noticeably lower closeness scores.

## 5.3 ABLATION ON SLOT GUIDANCE AND GMM REGULARIZATION

We ablate the influence of the *slot* guidance component $\nabla_{\mathbf{s}} \mathcal{L}_{\text{cf}}$, by turning off noise guidance (not updating $\epsilon_\theta$ to $\hat{\epsilon}_\theta$, see Eq 7 and varying the presence of the GMM regularizer and the slot update step size $\eta_s \in \{1.0, 0.3\}$. The first row in Table 2 reports the full model (slot+noise) for reference. *Slot-only + GMM* is robust to $\eta_s$ and remains close to the full model on all metrics. Crucially, locality remains high (ELS $\approx 71\%$), indicating that the slot signal is doing most of the heavy lifting for targeted edits. Removing the GMM regularizer degrades performance across the board and makes the method sensitive to $\eta_s$, consistent with off-manifold drift when slot updates are unconstrained. Against diffusion baselines, *slot-only + GMM* still outperforms L-DVCE, underscoring the importance of slot-level guidance. Adding noise guidance $\hat{\epsilon}_\theta$ provides a small but consistent boost to performance.

## 5.4 USER STUDY

Following DVCE (Augustin et al., 2022), we conducted a small user study with 20 participants. Each participant completed 8 tasks, where each task included 3 images (a query image alongside VOCCE and L-DVCE counterfactuals) and a textual description of the target modification. The participants were asked whether each counterfactual (1) had *meaningful changes*, (2) had *subtle changes*, and (3) had a *realistic appearance*. The aggregated results (for VOCCE / L-DVCE) are as follows:

Table 3: FFHQ-trained models on CelebA-HQ (means and relative change vs. multi-dataset training). Relative change in parentheses compares to models trained on *FFHQ+CelebA-HQ*. Bold percentages mark the best (least degradation or largest improvement) per metric.

| METHOD | CLOSENESS | | VALIDITY | | LOCALITY | REALISM |
|---|---|---|---|---|---|---|
| | $l_2 \downarrow$ | LPIPS $\downarrow$ | Succ. (%) $\uparrow$ | Fail. (%) $\downarrow$ | ELS (%) $\uparrow$ | sFID $\downarrow$ |
| VOCCE$_{t_c=250}$ | 52 **(+18%)** | 0.25 **(+15%)** | 86.5 **(0%)** | 16.1 **(+3%)** | 68.4 **(-4%)** | 42.5 **(+20%)** |
| L-DVCE$_{t_c=250}$ | 61 (+25%) | 0.34 (+23%) | 68.6 **(0%)** | 15.1 (+6%) | 51.7 ($-9\%$) | 47.3 **(+20%)** |

- **Meaningful:** 131/160 (82%) vs. 109/160 (68%)

- **Subtle:** 126/160 (79%) vs. 88/160 (55%)

- **Realistic:** 61/160 (38%) vs. 54/160 (34%)

These findings indicate that VOCCE generates more meaningful and subtle modifications compared to L-DVCE. While the realism scores are somewhat lower for both methods, the results suggest that future improvements could focus on preserving realism while maintaining the meaningfulness and subtlety of the edits.

## 5.5 GENERALIZATION ACROSS DATASETS

One of the main promises of object-centric learning is a better capacity for generalization (Greff et al., 2020). To evaluate this, we train both diffusion models on FFHQ exclusively and test them on CelebA-HQ. Table 3 reports both the absolute performance and relative change compared to the same models trained on both FFHQ and CelebA-HQ. VOCCE retains higher performance with smaller degradation in closeness (LPIPS) and realism (sFID), demonstrating its stronger ability to generalize to a new facial dataset. While L-DVCE also remains viable, it suffers more pronounced drops in closeness and realism, indicating less robust cross-dataset adaptation. These findings underscore the advantage of our object-centric diffusion approach in generating visual counterfactual explanations that transfer effectively across related datasets.

## 6 LIMITATIONS AND FUTURE WORK

As our approach is built on DDPMs and Slot Attention, it inherits all the limitations of these backbone models. A notable one is computation time: generating 64 counterfactuals using NVIDIA RTX A6000 GPU takes 29 seconds against 24 seconds for L-DVCE. Another major limitation is the challenge of applying modern object-centric learning methods to large and diverse datasets such as ImageNet (Deng et al., 2009), where the concept of discrete objectness becomes less clear. In future work, we plan to investigate ways to extend VOCCE to more complex image datasets and to other visual modalities like videos and 3D-scenes.

## 7 CONCLUSION

We proposed VOCCE – the first object-centric framework for generating visual counterfactual explanations that target meaningful, slot-level changes while preserving realism. By leveraging classifier guidance, gradient-based slot updates, and a GMM regularizer within a diffusion-based model, our approach achieves state-of-the-art performance on CelebA-HQ and synthetic ClevrTex, especially in terms of locality, closeness, and plausibility. Notably, VOCCE demonstrates stronger generalization than purely pixel-level diffusion baselines, exhibiting less performance degradation when trained on FFHQ and evaluated on CelebA-HQ. A user study further confirms the advantages of VOCCE: participants rated its explanations as more meaningful (82% vs. 68%), more subtle (79% vs. 55%), and more realistic (38% vs. 34%) compared to L-DVCE. Collectively, these findings highlight how object-centric diffusion enables more interpretable, realistic, and targeted visual counterfactuals, marking a significant step toward actionable explainability for complex vision models.

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
