# OpenReview forum: "Visual Object-Centric Counterfactual Explanations"
_ICLR.cc/2026/Conference — ICLR 2026 Conference Withdrawn Submission_

### Official Review · Reviewer_8sJi · 2025-10-29

**Soundness:** 2
**Presentation:** 2
**Contribution:** 2
**Rating:** 4
**Confidence:** 4

**Summary:**

To address the issue of existing methods creating unrealistic or entangled visual counterfactual explanations (CEs), this paper introduces a novel object-centric approach VOCCE. The method aims at decomposing images into distinct object-centric latent slots and uses a gradient-guided reverse diffusion process, constrained by a GMM regularizer, to achieve realistic edits that are localized to specific objects. To assess these claims, the paper presents a series of experiments, including quantitative comparisons and a user study.

**Strengths:**

S1. The paper considers an important problem of constraining the changes in CEs to some semantically meaningful subset of image pixels.

S2. The proposed method's novelty lies in combining latent slot diffusion with a GMM-based regularization and gradient-based slot updates within the diffusion reverse process conditioned on the classifier of interest.

S3. In addition to the standard quantitative evaluation, the authors conduct a user study to assess important properties of the generated CEs, and achieve optimistc results.

**Weaknesses:**

W1. My main concern relates to whether the primary premise of VOCCE is actually delivered. While I acknowledge its superiority in terms of the ELS metric, what are the actual guarantees that the resulting CE has changes only within a specific subset of the slots? From what I understand, the changes should be following the distribution of slots in the training data, but this doesn't mean that less significant slots won't be modified at all. Therefore, the changes might be more focused around regions that are more influential on the classifier, but also appear in less important regions, which contradicts locality. This can also be seen in Figure 2, e.g., columns 2, 5 and 6 for VOCCE include visible background changes.

W2. The authors properly mention recent state-of-the-art solutions like DiG-IN, ACE and ECED, but never actually include them in the quantitative evaluation. This directly contradicts the claim that VOCCE *yields significantly more interpretable and realistic counterfactuals compared to **state-of-the-art** baselines*. Moreover, I believe that the paper is also missing another relevant baseline [1], which also considers localized, semantically valid edits performed within predefined regions -- a concept that resembles slots used in VOCCE.

W3. An often overlooked issue in the evaluation of CEs is a lack of explanandum-related metrics. In this paper, the only such metric is the flip rate (validity). Many recent and prior works additionally include COUT [2], which focuses on classifier-related sparsity. Including this metric is important, as its role is largely orthogonal to those included in the authors' work.

W4. I acknowledge that the authors mentioned the challenge of VOCCE's applicability to larger and more diverse datasets as a limitation. I believe that such an example is crucial for a paper that puts object-level semantics at its core. At least an initial study into such an example should be provided, e.g., on BDD100k [3] or ImageNet [4].

W5. Missing ablation studies: what is the influence of $\gamma$, the number of Gaussian components and the step size for the slot update on the optimization procedure?

W6. Less significant but still important:

1. line 101: counterfactuals refer to images with specific characteristics edited, not conditioned on the classifier. Counterfactual explanations should be mentioned here instead.

2. No qualitative examples for the ClevrTex dataset.

3. It would be beneficial to include qualitative examples presenting the absolute difference between the original samples and their corresponding CEs. These would visually indicate the actual locality and object-centredness of the changes.


[1] Sobieski et al., Rethinking Visual Counterfactual Explanations Through Region Constraint, ICLR, 2025

[2] Khorram et al., Cycle-Consistent Counterfactuals by Latent Transformations, CVPR, 2022

[3] Yu et al., BDD100K: A Diverse Driving Dataset for Heterogeneous Multitask Learning, CVPR, 2020

[4] Deng et al., ImageNet: A large-scale hierarchical image database, CVPR, 2009

**Questions:**

Please refer to the Weaknesses above.

---

### Official Review · Reviewer_gb21 · 2025-10-31

**Soundness:** 3
**Presentation:** 1
**Contribution:** 2
**Rating:** 4
**Confidence:** 4

**Summary:**

The paper introduces **VOCCE**, a method for generating visual counterfactual explanations that act on *object-centric representations* rather than entire images. By decomposing scenes into object slots and modifying only those relevant to a target label change, the approach aims to produce more interpretable and localized counterfactuals. Experiments on datasets such as CLEVRTex, FFHQ, and CelebA-HQ compare VOCCE to existing diffusion-based counterfactual methods, reporting improved realism and faithfulness, along with a user study supporting better perceptual quality.

**Strengths:**

- **Comprehensive ablation study:** The paper includes an ablation analyzing the contribution of individual loss terms and components, which helps assess design choices.
- **User study included:** A small-scale human study comparing generated counterfactuals is a welcome addition, showing effort to evaluate beyond automatic metrics.
- **Potentially useful insight:** The approach could, in principle, enable explanations that modify only the relevant object in a scene, which may aid fine-grained interpretability.

**Weaknesses:**

The main limitation of the paper is **how it is written and organized**, which makes it unnecessarily difficult to follow and evaluate.
1. **Unclear motivation and intuition**
   The paper lacks an intuitive justification for why *object-centric explanations* are particularly needed or advantageous. It would help to include a concrete example (e.g., modifying only one object’s attribute) showing when such explanations outperform traditional pixel- or region-based methods.

2. **Poor presentation and confusing figures.**
   - Figure 1 is overly complex and difficult to interpret; subfigure (a) references terms that do not appear in the diagram, and subfigure (b) is too dense, without clear explanation.
   - The overall architecture description is fragmented: it is unclear which components are pretrained, fine-tuned, or frozen, and when the classifier is first introduced versus reused in later sections (3.1 vs 3.2).

3. **Loss definitions are confusing.**
   - The slot loss is re-introduced multiple times (Sections 3.3 and 3.4), and it remains ambiguous which terms constitute the final objective versus auxiliary ones.
   - The training setup (e.g., what is pretrained on FFHQ, what is optimized jointly) is not clearly specified.

4. **Experimental setup**
   - Only two datasets are used (CLEVRTex and FFHQ), and there is no qualitative analysis on CLEVRTex. Including additional datasets or broader discussion would strengthen the empirical claims.
   - It remains unclear how the authors chose the set for sFID evaluation.

5. **Quantitative results.**
   - The paper does not report results for edits involving multiple labels or multiple object changes simultaneously, which could be a selling point of the approach.
   - The user study compares only with L-DVCE and omits other relevant baselines; no statistical significance tests are provided.
   - Same for the generalization experiment

Overall, the **writing quality and structure** significantly hinder readability and make it difficult to understand the model, experiments, and contributions.

**Questions:**

1. What specific advantage do *object-centric explanations* provide over region-based or pixel-level counterfactuals? Could you illustrate a concrete scenario where your approach succeeds but standard ones fail?
2. Please clarify the training procedure: which models are pretrained, frozen, or fine-tuned? What is the purpose of FFHQ pretraining?
3. How is the “slot loss” finally combined with classifier and reconstruction losses? A full loss equation would help.
4. Why are only two datasets used, and why is there no qualitative analysis on CLEVRTex? Would the approach generalize to other datasets or object domains?
5. How were sets selected for computing sFID?
6. In the user study, why only compare with \(L_{\mathrm{DVCE}}\)? Were significance tests (e.g., paired t-test, bootstrap) performed?
7. Could you simplify or redesign Figure 1 for readability and ensure all terms used in the caption appear in the diagram?

---

### Official Review · Reviewer_UQUx · 2025-11-01

**Soundness:** 2
**Presentation:** 2
**Contribution:** 3
**Rating:** 2
**Confidence:** 4

**Summary:**

This work proposes an object-centric method for generating realistic and interpretable visual counterfactual explanations. Unlike prior pixel-level or holistic latent approaches that produce entangled and unclear edits, which are done by firstly decomposing images into object-centric latent slots, isolating meaningful components and later use the model gradients to guide a reverse diffusion process for generating counterfactuals, While appling a GMM-based regularizer to ensure generated counterfactuals to prevent slot drifts.

**Strengths:**

- The use of OCL for counterfactual generation is interesting, enabling swapping of entire object rather than some abstract parts, mainting human understandability to an extent

- The idea of GMM regularisation is interesting and novel

- The paper is well written and easy to follow to a large extent

**Weaknesses:**

- For the GMM prior generation, there exist a few methods, like: (1) SlotVAE-like approach with learnable GMM priors and learn the parameters via training, (2) Probabilistic Slot Attention (PSA) like approach using an aggregate posterior to create an optimal prior, (3) Or the approach the authors have considered using the latents of the entire datasets and manually fitting the GMM, the implications of all three methods or atleast the discussion on three methods would add value to the paper.

- The details of GMM fitting are missing. Do you consider latents from the entire dataset? Or subset? If a subset, how do you select these samples? Discuss the tractability and complexity of such a method.

- More recent works on realistic counterfactual generations are not cited: High Fidelity Image Counterfactuals with Probabilistic Causal Models, Diffusion Counterfactual Generation with Semantic Abduction, and others, as they achieve more realistic-looking counterfactuals, which the authors claim do not exist

- Details about the human survey aren’t clear

- OCL results in isolation are not discussed in qualitative nor quantitative form

**Questions:**

- The use of OCL is limited to GMM regularisation. I’m curious to hear the author's thoughts on guiding diffusion models to generate appropriate slots themself directly.

- Would like to know if the following OCL approach helped in the case of the FFHQ dataset? based on the types of edits, its bit hard to gauge, as OCL methods are not capable of capturing abstract concepts like young, smile, and so on.

---

### Official Review · Reviewer_rfdB · 2025-11-01

**Soundness:** 2
**Presentation:** 2
**Contribution:** 2
**Rating:** 4
**Confidence:** 3

**Summary:**

This paper proposes VOCCE (Visual Object-Centric Counterfactual Explanations), a diffusion-based method designed to generate realistic and interpretable counterfactual images. The method employs classifier-guided diffusion to steer samples toward a target label, gradient-based slot updates to refine object-level changes, and a Gaussian Mixture Model (GMM) prior to maintain plausibility. Experiments on ClevrTex, FFHQ, and CelebA-HQ show improved closeness, realism, and locality over prior diffusion-based counterfactual methods (DVCE, DiME, FastDiME). A user study further supports that VOCCE produces more meaningful and subtle edits, and cross-dataset tests indicate strong generalization of object-centric representations.

**Strengths:**

- The paper is motivated well and I enjoy reading the introduction part.
- It claims to be the first object-centric framework for generating visual counterfactual explanations.
- The method is tested on three datasets, showing consistent quantitative improvements over baselines.
- The visual results in figure 1 look realistic and with good identity preservation.
- They show generalization of the proposed method by training on FFHQ and tested on CelebA-HQ.
- They perform human user study which confirms benefit of their method over the baseline.

**Weaknesses:**

- The main difference from prior diffusion-based counterfactual methods appears to be the introduction of slot attention for object-centric representations. Yet there are no experiments focusing on exploring how each slot captures different object and how that help with the counterfactual generation.
- It is hard to follow the methodology. For example, it is not clear how different components of the method is trained. Are they trained separately or jointly? The methodology section needs to be improved. And maybe an algorithm summarizing the method step by step would be helpful for clarity. Unfortunately, in my opinion, figure 1 is confusing and not very helpful for understanding.
- One key contribution of this paper is that it operates on slot-based representations, where “each slot encodes a distinct object”. However, this is not experimentally verified in this paper. I would expect some experiments where they visualize which object a slot captures. And how this can help with the counterfactual quality. For example, for mouth open 0->1, is there a slot that represents mouth? When how does that slot representation changes during the counterfactual generation process? And how do other slots change?
- The paper focuses on generating visual counterfactual explanations, yet it presents only one figure with qualitative results on CelebA-HQ. Since the main contribution is visual interpretability, it would be helpful to include more qualitative examples, ideally across multiple datasets and target attributes, to better demonstrate the quality of the generated counterfactuals.
- It is unclear whether the same classifier is used both to guide the counterfactual generation and to evaluate validity. If so, the reported success rate may be biased, since the classifier’s own errors or overconfidence could make invalid counterfactuals appear “successful.” It would be important to clarify this setup and, if possible, evaluate validity using an independent or held-out classifier to ensure that the measured success reflects genuine semantic changes rather than the guiding model’s bias.

**Questions:**

- Can you provide more clear description of the methodology, and maybe provide an algorithm summarizing the method step by step.
- In this paper, “counterfactual” is defined as the minimally modified image that flips a classifier’s prediction. While this aligns with the machine-learning literature on counterfactual explanations (e.g., Wachter et al., 2017; Goyal et al., 2019), it differs from the causal-inference definition by Judea Pearl, where counterfactuals are potential outcomes under explicit interventions in a causal model. Pearl’s formulation requires structural equations and causal mechanisms, not just classifier decision boundaries. Thus, the proposed method generates decision-boundary counterfactuals rather than causal counterfactuals, and this distinction should be clarified to avoid confusion in terminology.
- It would also be valuable to discuss the difference between this line of work and causal-model-based counterfactual generation methods that explicitly incorporate Pearl’s framework (e.g., Deep Structural Causal Models, Pawlowski et al., 2020). Can such causal generative models also serve as counterfactual explanation tools, and how do their assumptions or objectives differ from the classifier-driven approaches explored here?
- Can you provide experiments where you can show how each slot capture objects? And how different slot changes for different attribute counterfactuals?
- Can you provide more visual results? For the other two datasets and for the ablation studies.
- Did you use the same classifier for guiding the counterfactual generation and to evaluate validity? If so, the reported success rate can be biased.
- Did you consider the issue of attribute entanglement when evaluating counterfactuals? If two attributes are correlated or causally linked (e.g., Smiling and Mouth open in CelebA-HQ), flipping one may unintentionally affect the other, making it unclear whether the change is truly isolated. It would be useful to discuss how your method handles such entangled attributes, whether through disentanglement in the slot space, targeted conditioning, or evaluation metrics that account for correlated attributes.
- In section 6, the authors wrote "our approach is built on DDPM", but in 4.2, they wrote "we use DDIM sampling". So do you use DDIM or DDPM?
- When generating images, did you first get a noisy version of z and then apply classifier guidance during the reverse process? If so, I think you did use DDIM, right?

---

### Note · Authors · 2025-11-23

I have read and agree with the venue's withdrawal policy on behalf of myself and my co-authors.